# Formative Research on HPV Vaccine Acceptance among Health Workers, Teachers, Parents, and Social Influencers in Uzbekistan

**DOI:** 10.3390/vaccines11040754

**Published:** 2023-03-29

**Authors:** Sahil Khan Warsi, Siff Malue Nielsen, Barbara A. K. Franklin, Shukhrat Abdullaev, Dilfuza Ruzmetova, Ravshan Raimjanov, Khalida Nagiyeva, Kamola Safaeva

**Affiliations:** 1Consultant, World Health Organization Regional Office for Europe, DK-2100 Copenhagen Ø, Denmark; 2Vaccine-Preventable Diseases and Immunization Programme, World Health Organization Regional Office for Europe, DK-2100 Copenhagen Ø, Denmark; 3All One Communication, San Diego, CA 92117, USA; 4ITA FACT, Tashkent 100000, Uzbekistan; 5World Health Organization Country Office in Uzbekistan, Tashkent 100100, Uzbekistan

**Keywords:** human papillomavirus, vaccine introduction, barriers, vaccine acceptance, behaviour change, Uzbekistan

## Abstract

Human papillomavirus (HPV) vaccines effectively prevent cervical cancer, most of which results from undetected long-term HPV infection. HPV vaccine introduction is particularly sensitive and complicated given widespread misinformation and vaccination of young girls before their sexual debut. Research has examined HPV vaccine introduction in lower- and middle-income countries (LMICs), but almost no studies attend to HPV vaccine attitudes in central Asian countries. This article describes the results of a qualitative formative research study to develop an HPV vaccine introduction communication plan in Uzbekistan. Data collection and analysis were designed using the Capability, Opportunity, and Motivation for Behaviour change (COM-B) mode for understanding health behaviours. This research was carried out with health workers, parents, grandparents, teachers, and other social influencers in urban, semi-urban, and rural sites. Information was collected using focus group discussions (FGDs) and semi-structured in-depth interviews (IDIs), and data in the form of participants’ words, statements, and ideas were thematically analysed to identify COM-B barriers and drivers for each target group’s HPV vaccine-related behaviour. Represented through exemplary quotations, findings were used to inform the development of the HPV vaccine introduction communication plan. Capability findings indicated that participants understood cervical cancer was a national health issue, but HPV and HPV vaccine knowledge was limited among non-health professionals, some nurses, and rural health workers. Results on an opportunity for accepting the HPV vaccine showed most participants would do so if they had access to credible information on vaccine safety and evidence. Regarding motivation, all participant groups voiced concern about the potential effects on young girls’ future fertility. Echoing global research, the study results highlighted that trust in health workers and the government as health-related information sources and collaboration among schools, municipalities, and polyclinics could support potential vaccine acceptance and uptake. Resource constraints precluded including vaccine target-aged girls in research and additional field sites. Participants represented diverse social and economic backgrounds reflective of the country context, and the communication plan developed using research insights contributed to the Ministry of Health (MoH) of the Republic of Uzbekistan HPV vaccine introduction efforts that saw high first dose uptake.

## 1. Introduction

Most HPV infections are spread through sexual contact and are asymptomatic. Persistent genital HPV infection can cause cervical cancer: the fourth most common cancer in women worldwide [1]. Approximately 85% of global cervical cancer cases occur in lower- and middle-income countries (LMICs), with almost all cases linked to genital HPV infection [2]. Uzbekistan is an LMIC where cervical cancer is the second most common cancer in women, and approximately 1608 new cases are diagnosed annually [3].

HPV vaccines are estimated to have one of the highest per-person impacts on mortality [4]. They are efficacious in preventing infection with HPV strains responsible for approximately 70% of global cervical cancer cases [5]. These vaccines are mainly administered to adolescent girls 9–18, with vaccination recommended before their sexual debut. Despite the high discrepancy in uptake between high-income countries (32.1%) and LMICs (0.1–7.2%), limited research is available on HPV vaccine introduction in LMIC contexts [6,7,8]. While some research has been conducted on cervical cancer and HPV in Central Asia, almost no research is available on the HPV vaccine introduction in this region [9,10,11].

The Ministry of Health of the Republic of Uzbekistan (MoH) planned the first phase of a nationwide HPV vaccine introduction, to be conducted using school-based vaccination of young girls in October 2019. The global experience illustrates that school-based HPV vaccination programmes can be compromised by a range of factors including parents’ vaccine hesitancy, teachers’ and health workers’ misinformation and disengagement, and negative media coverage [7,12,13,14,15,16]. The multiple actors and factors affecting HPV vaccine uptake render its introduction particularly complex compared to the more traditional infant and child immunization [5,17].

Ahead of introducing the new vaccine, the MoH aimed to prepare an introduction communication plan, with technical support from the World Health Organization (WHO) Regional Office for Europe. With five months to develop the plan, the WHO Tailoring Immunization Programmes (TIPs) three-stage approach to planning, research, and intervention development was adapted to design a study focussed specifically on HPV vaccine introduction [18]. This article reports the findings of a rapid qualitative study conducted in the second stage to understand barriers and drivers to general and HPV vaccination among key target groups in Uzbekistan. The process and outcomes of undertaking such rapid, theory-based qualitative research will be developed in a future publication.

## 2. Materials and Methods

### 2.1. Study Design

In the first stage of the TIP process, an iterative process of literature review and stakeholder consultation was used to identify appropriate target groups, research focus, research methods, and sample sizes. Stakeholders consulted included MoH representatives, State Sanitary-Epidemiological Service representatives, and WHO Country Office technical staff. Academic databases and grey literature were searched for data on HPV vaccine introduction and interventions, particularly in LMICs. A research protocol for the three-month qualitative study, including discussion guides based on the themes identified was developed and submitted for ethical approval.

The data collection tools and thematic analysis approach were guided by the Capability, Opportunity, and Motivation for Behaviour change (COM-B) model [19]. Developed on a review of nineteen existing behaviour change frameworks, the COM-B model asserts specific public health behaviours are affected by three interrelated factors of individuals’ capability, opportunity, and motivation to perform the behaviour [20,21]. Table 1 provides descriptions and examples of the COM-B factors.

Data were collected across four sites. Districts A and B selected in Tashkent City, respectively, represented lower- and upper-socio-economically stratified areas. District C in the Tashkent region was selected as a semi-urban site inhabited by internal migrants from across Uzbekistan, and District D represented a socio-economically and ethnically more homogenous, rural population. Sample sizes were determined using the concept of ‘information power’ for qualitative research, to capture pertinent social issues, relationships, and dynamics within target groups, considering study aims, sample specificity, use of the theoretical framework, discussion quality, and analysis strategy [22].

### 2.2. Research Participants

Participants from four target groups were included in the research. The first group, parents and grandparents of girls aged 8–10, are key actors in children’s health decision-making in Uzbekistan. Only parents or grandparents of girls aged 8–10 years were included. Secondly, health workers (GPs, patronage nurses), were included as they are considered trusted sources of information in Uzbekistan who influence parents’ vaccination decisions. Only GPs and nurses involved in children’s vaccination from the target districts were included. The third target group consisted of teachers, as school-based vaccine introduction was planned using teacher mobilisation of parents and students. Only teachers of girls aged 8–10 were included. The fourth target group consisted of social influencers who directly or indirectly impact parents’ decision-making. These included religious officials, mahalla (neighbourhood committee) leaders, school principals, gynaecologists, oncologists, social media influencers, and key national-level health representatives. Social influencers were included based on their work in areas connected to vaccination or women’s and girls’ health.

Research participants were purposively sampled. Health workers, parents, and grandparents were identified from local polyclinics, and teachers were selected from elementary schools in each site. Social influencers were purposively selected by researchers’ professional networks using snowball sampling. Bloggers with a social media presence and history covering health issues were identified and contacted by the communication team at the WHO Country Office in Uzbekistan.

While research teams had planned to include parents and grandparents to represent both sexes and different social backgrounds, unforeseen time and resource constraints resulted in research only being conducted with mothers and grandmothers. Thus, feasible, contextual information on perceptions of men’s behaviour, knowledge, etc., was garnered from participants across other target groups.

### 2.3. Data Collection

Data were collected using audio recordings, written notes, and observations of FGDs and IDIs were conducted with participants.

Research guides were initially developed in English, translated into Russian and Uzbek and reviewed line-by-line to ensure correspondence with the original English versions. The guides covered the following issues identified in the study planning phase as affecting target groups’ behaviour in promoting or accepting the HPV vaccine:Knowledge of cervical cancer, HPV, or the vaccine (including information sources and ability to communicate information);Access to vaccination services or resources (time, money, etc.) affecting access;Systemic or societal factors influencing the decision to vaccinate young girls, including the language used to evaluate or discuss vaccination.

All FGDs and IDIs were led by a moderator and observed by a note-taker who recorded observations that were not captured in audio transcripts, e.g., group dynamics, tone, or body language, as well as two WHO researchers accompanied by a simultaneous English interpreter. Activities were audio recorded and transcribed verbatim in the original language. The transcripts and note-taker’s observations were translated into English prior to analysis.

### 2.4. Ethical Considerations

Informed consent was obtained from all subjects involved in this study. Prior to inclusion, all participants provided written consent in either Uzbek or Russian. Since this research aimed to gauge participants’ knowledge of vaccination and HPV, they were informed that the discussions would cover vaccination topics, but the HPV vaccine introduction was introduced only during the research activities.

This study was conducted in accordance with the Declaration of Helsinki and approved by the Ethics Committee of The Ministry of Health of the Republic of Uzbekistan (protocol number 2/34-1018, approved 26 February 2019).

### 2.5. Data Analysis

Information was coded and analysed thematically by hand using a deductive coding framework [23] to identify (i) trends, drivers, and barriers organized by COM-B factor for each target group regarding general vaccination and HPV vaccine introduction (both self-reported by individuals within a target group or expressed by participants in other target groups); (ii) participant suggestions to improve general vaccine uptake or promote HPV vaccine acceptance; and (iii) recommendations on messages and media for HPV vaccine introduction communication.

Target group behaviours explored under the COM-B codes included (i) parents’ and grandparents’ decision to vaccinate young daughters/granddaughters, (ii) health workers encouraging and administering vaccination to young girls, (iii) social influencers (including teachers) encouraging vaccination of young girls; and (iv) national level key informant perceptions of context and commentary relating to other target groups’ behaviours.

To ensure standardised coding, the researchers analysing the data developed an initial coding framework and independently coded the same three FGD transcripts from different target groups. Finding over 75% agreement in coding, they agreed on a shared approach before proceeding independently. Any emerging codes were discussed jointly before inclusion in the framework. The findings of each transcript were recorded in individual tables organized by code with example quotations. Findings from each table were also considered against notes available for each research activity to glean any additional contextual information. Finally, the individual transcript summaries were aggregated into four tables summarizing COM-B findings on vaccination-related behaviour for each target group.

## 3. Results

A total of 22 FGDs and 10 IDIs were conducted with 164 participants. Findings on barriers to the performance of behaviours largely cut across target groups and sites, with some minor differences. Results are provided in the sections below by COM-B factor. An overview of participants and FGDs and IDIs conducted across field sites is provided in Table 2.

### 3.1. Capability Factor Findings

Almost all participants understood the role of vaccination in strengthening immunity. A recurring issue among health workers was confidence in communication with vaccine-hesitant patients, especially with less educated patients or those with religiously based concerns.


*“Why should we speak to [people refusing vaccination]? For example, in educated families, medical culture is accepted, access to information isn’t a problem … It is very difficult to work with religious or uneducated families. We … explain everything, but they do what they want in the end. Working with them is a little challenging”.*

*—Doctor, District B, Tashkent*


All health workers understood the connection between HPV and cervical cancer. Those with prior training conducted under earlier HPV vaccine introduction planning were markedly well-informed on HPV, while health workers without training were reservedly supportive of the vaccine. Rural GPs without training had the least information and highest misunderstandings of HPV transmission, e.g., prevention through monogamy or genetic predisposition. A few nurses in the more affluent Tashkent City district also held reservations about an HPV vaccine, reflecting knowledge gaps on immunization.


*Moderator: What are your concerns [on recommending the HPV vaccine to family]?*



*Participant 1: Infertility … they might come to us later saying ‘you made [my child] take it’.*



*Participant 2: There were some reactions after the [Diphtheria-Tetanus-Pertussis] vaccine, such as a fever … polio leads to paralysis … since [HPV vaccine is] against cancer, would it cause cancer?*

*—Patronage Nurses, District B, Tashkent*


Data suggested a difference in urban and rural parents’ level of vaccine hesitancy, though misinformation, unexpectedly, did not appear to be a widespread issue. Some participants across all groups indicated some parents might refuse specific vaccines such as the pentavalent vaccine due to misinformation on potential side effects, though key informants from the MoH suggested misinformation on vaccination was more prevalent among urban parents. Urban health workers and teachers were more likely to report experience with parents refusing vaccines than their rural counterparts, both for religious and non-religious reasons.


*“…[urban mothers share] negative information on Telegram … They have their own groups for mothers and they throw in the information to exchange with each other: somewhere, someone got sick, somewhere, someone has died from vaccination, etc… I believe that incorrect information is the main reason for refusing from vaccination”.*

*—Official, Mother and Child Care, MoH*



*“There are many people from provinces [here] … They never refuse immunization … they would stay at the hospital with their two-month old babies and get all possible vaccinations if allowed … Urbanites are completely different …”*

*—Doctor, District A, Tashkent*



*“We talk about [vaccination] at teacher-parent meetings, because many parents … are against vaccination … [and] write waiver notes”.*

*—Teacher, District B, Tashkent*


Vaccine hesitancy stemming from knowledge gaps on safety and international regulation of vaccines was found among both urban and rural mothers, grandmothers, and teachers. Except for health workers, few research participants were aware of HPV, its relation to cervical cancer, and the HPV vaccine. Only a few teachers in Tashkent had heard of the HPV vaccine via a news item that had run days before this research was conducted. Several participants across all sites expressed misconceptions that HPV infection in women resulted from not addressing a cold or that it could be cured by delivering another child.

Rural GPs without prior HPV vaccine training and teachers were the most sceptical of the HPV vaccine, indicating they needed more information to encourage vaccination. Across all groups, individuals requested detailed information on the HPV vaccine’s evidence base and safety, preferably with examples from Uzbekistan. Teachers and health workers also requested guidance on effective communication with parents on HPV vaccination.


*“… previously they didn’t explain anything to us. When the [HPV] vaccine was [piloted], they said: ‘this this the vaccine, it is against this, do it—quickly find girls. We barely knew anything ourselves and could not explain it well”.*

*—Doctor, District D, Tashkent Region*


### 3.2. Opportunity Factor Findings

The majority of participants across all groups reported trusting and wanting information on vaccination and other health issues from health workers or the MoH. Health workers also felt patients saw them as trusted sources of information. Mahallas, women’s committees, schools, religious leaders, and polyclinics indicated they collaborated to provide health education to various constituents verbally and electronically. Teachers, school principals, and ministry officials described how, as part of the national curriculum, teachers discuss general vaccination with parents, cover vaccination-related topics in class work, and encourage students to discuss vaccination with parents.


*“Through groups of our little girls in nursery schools we try to cover not only issues related to education … but also health with their parents and grandmothers. We invite medical nurses and doctors to explain hygiene and medical issues”.*

*—Women’s Committee Representative, District D, Tashkent*



*“Every Friday there is pedagogic hour [in schools], where vaccination might be discussed … there are separate parent groups and channels in Telegram, where we can organize explanatory works [for parents on vaccination]”.*

*—Ministry of Education Representative, Tashkent*


While almost all health workers stated they relied on the MoH for information on vaccines, some nurses mentioned turning to internet sites or search engines. Except for health workers, participants across the other groups and sites indicated relying on vaccination information from family, friends, neighbours, or others with first-hand experience. Such information might be communicated in person or shared using social media. Social media was also used to circulate articles or videos related to vaccination, though less so among rural participants. Several urban mothers reported checking health worker information online before making vaccination decisions. Urban participants felt vaccination decisions are made by mothers and fathers jointly, while non-urban parents felt fathers or grandmothers played a stronger role.


*“I surf the Internet to find information on vaccines …I’ve learned for myself [to check what health workers have told], and then gotten the vaccine … With the exception of hepatitis A, I got other vaccines [for my kids] on time”.*

*—Mother, District B, Tashkent*



*“I might ask my older sister-in-law [about health questions] if I don’t understand … or doubt doctors … she explains it to me because she knows”.*

*—Mother, District C, Tashkent Region*


In terms of discussions that promote the HPV vaccine among parents, health workers and teachers felt social attitudes would be a barrier to discussing the HPV vaccine one-on-one with parents. Several health workers felt discussing issues touching on reproductive health with parents of young daughters would be difficult and could lead to misunderstandings or suspicion of the vaccine. Most teachers felt it was inappropriate for them to bring up the issue of HPV with parents of young girls when discussing the vaccine and felt it would be better received from a health worker.


*“[parents] might think: ‘they only started promoting [HPV vaccine], who knows what it might result in … After all, In Uzbek culture one has to protect girls”.*

*—Nurses, District C, Tashkent Region*



*“… we have Uzbek cultural ways, there is some self-consciousness [in talking about the womb] in the presence of men… it may be a little inconvenient to talk about [the new vaccine]”.*

*—Teacher, District B, Tashkent*



*“Doctors just have to [tell us], ‘We guarantee [the vaccine is safe], trust us’ … After this, if I were to be held accountable, I could explain [information on the vaccine].”*

*—Teacher, District D, Tashkent Region*


Study participants who were in the capital and capital region did not report any structural issues in access to health facilities, though a few urban mothers and grandmothers complained about long wait times on immunization days. Health workers similarly indicated high patient loads, and a few felt they were inadequately remunerated.

### 3.3. Motivation Factor Findings

Most participants supported HPV vaccine introduction even with the caveat of requiring more information on its safety. Mothers, grandmothers, teachers, and several nurses stated that, as with other vaccinations, they would first wait to see how the daughters of others fared before deciding to vaccinate their own. Their caution was grounded in a concern shared across target groups of any potential negative effects of HPV vaccination on girls’ reproductive health and future fertility.


*“It would be difficult to recommend [the HPV vaccine] before you know … [if it] is studied thoroughly, has no side effects … and [you see] a person who has taken this vaccine … Girls are a delicate matter … One might fear harming their fertility”.*

*—Grandmother, District B, Tashkent*



*“We [mothers] ask each other … ‘how was it with vaccinating your children?’ After [others] say it went well, we allow vaccination of our children”.*

*—Mother, District A, Tashkent*


The concern for girls’ future fertility was mentioned by teachers and health workers as a factor that would negatively impact their decision to encourage vaccination, as they did not want to be held responsible by parents for any negative outcomes. On the other hand, the same concern around fertility was seen as a positive factor by some participants who considered the longer-term impact of preventing cervical cancer for girls who would one day become mothers. Ultimately, however, all participants cited confidence in their knowledge and understanding of the vaccine’s safety and evidence base as a major consideration in their supporting vaccination.


*“[The HPV vaccine] is necessary … We marry off our daughters but … We need to protect our girls from [HPV and cervical cancer] … Even if her husband catches the disease, our girl will not … be infected.*

*—Grandmother, District C, Tashkent Region*



*“My child should not have to say in the future that she became infertile ‘because my mother got me vaccinated with this vaccine’. There must be 100% certainty [of HPV vaccine safety] to avoid any harmful consequences”.*

*—Mother, District B, Tashkent*


## 4. Discussion

Effective HPV vaccine introduction interventions are theory-based and informed by formative research [24,25]. The study presented in this article was conducted as formative research to develop the 2019 Uzbekistan HPV vaccine introduction communication plan. COM-B theory was used to inform data collection, analysis, and subsequent intervention development.

The main capability-factor findings indicated that, despite understanding that vaccination strengthens immunity, knowledge and skill gaps exist for health workers and the public regarding HPV and cervical cancer, the HPV vaccine, and effective communication on the HPV vaccine. Opportunity-factor findings showed social norms could hinder discussing the HPV vaccine with parents, but high trust in the MoH and health works as health information sources, as well as local-level cross-sector partnerships, could facilitate efforts to inform the public. Regarding motivation, the potential effect of the HPV vaccine on girls’ future fertility was cited by participants as a primary concern that would need to be addressed using evidence of HPV vaccine safety.

This study presents the first behaviour theory-based exploration of HPV vaccine acceptance in the central Asian region. Most research on effective HPV vaccine introduction targeting adolescent girls has been conducted in higher-income countries [17,26,27]. In recent years, more studies have been conducted in LMICs, but almost no research on this issue is yet available for the Central Asian Region [9,28,29,30]. The findings of this study agree with global research on HPV vaccine introduction, as discussed below, which found that similar issues exist across countries, with contextually specific expressions. This qualitative formative research provides information on social dynamics, concerns, and discourses around HPV vaccination that affected HPV vaccination acceptance by different actors in Uzbekistan.

The findings of this study suggested a need to strengthen knowledge and understanding of HPV among health workers—particularly nurses and rural GPs—and teachers, as well as to develop the communication skills of these groups and confidence in recommending the HPV vaccination to parents of young girls. This finding echoes global research that indicates addressing concerns and strengthening the capacity of health workers, and teachers in school-based interventions, is effective in addressing barriers to vaccination by creating confidence in the vaccine, increasing recommendations for vaccination, and providing adequate skills to allay parents’ vaccination fears [31,32,33,34].

Research participants across all groups were found to be generally supportive of the new HPV vaccine introduction, though a majority highlighted the need for clear, accessible, and credible information on the evidence base and safety of the HPV vaccine in order to be confident in their support for an HPV vaccine. Misinformation on vaccines did not appear to be widespread among research participants, though there was some indication of vaccine hesitancy among urban parents and teachers, possibly spread over social media. This situation differed markedly from other high-income and LMIC contexts, where even before the COVID-19 pandemic, misinformation spread over media contributed to HPV vaccine hesitancy, decreased uptake, or even government suspension of HPV vaccination [35,36,37,38]. The finding of low levels of misinformation is not surprising, considering that, at the time of this research, Internet coverage and access remained out of reach for many in the country outside the capital, and it was only after fieldwork ended that many social media sites were unblocked across the country [39,40].

Knowledge of HPV or HPV vaccination for study participants outside the health system was limited, though awareness of cervical cancer was high. Health workers, neighbourhood committee members, and religious leaders, in particular, cited cervical cancer as a specific problem they would like to address within their communities. Regarding introduction of the HPV vaccine for young girls, the primary anxieties of participants across all target groups centred around any potential negative effects on girls’ future fertility. Echoing research from other contexts, study participants’ orientation to protecting girls’ fertility appeared as both a driver and barrier to HPV vaccine uptake, depending on factors such as availability of credible information, social propriety of discussing girls’ sexual health in mixed-gender settings, and confidence in one’s knowledge of HPV [7,28,41,42,43,44,45]. Research from eastern Europe and the Caucuses similarly advises that concerns around girls’ fertility can be effectively addressed by emphasizing the safety of the HPV vaccine and presenting it as a way to protect young girls’ future health [29,46].

Aside from the barriers, this formative research also found two drivers that could facilitate HPV vaccine acceptability and uptake. All participants indicated trust in health workers or the MoH as credible sources of medical information on HPV and vaccination. The case in Uzbekistan was fortunate, as other countries’ experiences show a lack of trust in governmental sources of information can be a barrier to HPV vaccine acceptance, especially in the case of introductions targeting young girls [28,29,47]. Additionally, schools, governmental institutions, and health facilities reported pre-existing and successful collaboration on providing health education to young girls, parents, and their communities. Such collaboration would allow for better coordination among various governmental bodies and institutions in implementing the HPV vaccine introduction in the country. Again, the situation in Uzbekistan was unique even in the region, where health and social welfare services were administered at the most local level with government collaboration and oversight [48,49].

The findings of this study were in line with broader research on HPV vaccine introduction and provided a missing view of the Uzbekistan context; however, some limitations can be noted. As a piece of applied qualitative research with a relatively small, purposively selected sample size, participants were selected as feasibly as possible to represent diverse socioeconomic backgrounds, though greater diversity could have been attained with geographically more widely spread data collection sites. Additionally, young girls, the potential targets of the introduction, were not included due to time and resource limitations. Nonetheless, the formative research results were implemented to design a national HPV vaccine introduction communication plan that would see 92% of 10-year-old girls in the country vaccinated by the end of 2019 [50]. Finally, this study was conducted prior to the global COVID-19 pandemic. Since then, multiple factors have affected vaccine confidence and uptake globally and regionally, and new research might be needed to ascertain if and how current attitudes to the HPV vaccine might have changed [51,52,53].

## 5. Conclusions

Using the COM-B theoretical framework to plan and conduct research, this study identified individual and contextual factors affecting HPV vaccine acceptance in Uzbekistan. Findings from this qualitative research informed the development of the 2019 Uzbekistan HPV vaccine introduction communication plan. The research findings highlighted factors contributing to vaccine acceptance identified in similar research from both high-income and LMIC contexts. Equipping all levels of health workers and teachers with not just information to boost their confidence in HPV vaccine safety and need but also in strengthening their skills to effectively communicate on HPV vaccine was a primary finding. As in other contexts, the need for accessible and credible information on vaccine safety, as well as on HPV and cervical cancer, was stressed by both the public and health workers. Concern for the HPV vaccine’s potential effects on fertility was found to be both a barrier to acceptance as well as a potential driver if vaccine safety were credibly presented. A low level of misinformation on vaccination was observed, potentially related to low Internet and social media coverage at the time, and cross-sector collaboration on health awareness and administration was identified as a strong driver supporting vaccine acceptance.

## Figures and Tables

**Table 1 vaccines-11-00754-t001:** COM-B Factors with Examples.

COM-B Factors	Description	Examples
**Capability**	Focussed on Individuals’: Knowledge Skills or confidence in one’s skills Resilience, strength, or stamina to engage in the necessary mental processes or physically perform the behaviour	Individuals’ knowledge of vaccination Health workers’ ability to communicate importance of vaccination
**Motivation**	Focussed on Individuals’: Attitudes, perceptions, risk assessment Intentions Values, beliefs Emotions, impulses, feelings	Parents considering potential outcomes or waiting to observe the response of other parents’ children to vaccination before deciding to vaccinate Fear or distrust of the health system and vaccination Not wanting one’s children to feel pain from vaccination
**Opportunity**	Attending to Physical or Social Context: Access, affordability, availability of resources Convenience, appropriateness, or affordances for behaviour Social–cultural cues, norms, influences Structural frameworks (legislation, organisations, rights, etc.)	Cues to action for health workers to advise about vaccination Scheduling to help health workers adequately attend to patients Access to polyclinics Social norms or family/friends’ influence on vaccination decisions

**Table 2 vaccines-11-00754-t002:** Research Participants by Activity across Sites.

		District A Urban, Lower-Middle Class	District B Urban, Affluent	District C Semi-Urban, Ethnically Diverse	District D Rural, Ethnically Homogenous	Total
**Mothers (FGD)**		n = 8 (8F)	n = 10 (10F)	n = 7 (7F)	n = 8 (8F)	33 (33F)
**Grandmothers (FGD)**		0	n = 9 (9F)	n = 9 (9F)	0	18 (18F)
**General Practitioners (FGD)**		n = 9 (9F)	n = 10 (9F, 1M)	n = 6 (5F, 1M)	n = 7 (7F)	32 (30F, 2M)
**Patronage Nurses (FGD)**		0	n = 10 (10F)	n = 8 (8F)	0	18 (18F)
**Teachers (FGD)**		n = 8 (8F)	n = 8 (8F)	n = 9 (9F)	n = 8 (8F)	32 (32F)
**Social Influencers**	**IDI**	School Principal n = 1 (1F)	Gynaecologist n = 1 (1F)	Mahalla Representative, n = 1 (1F)	Women’s Committee, n = 1 (1F) Religious Leader, n = 1 (1F) School Principal n = 1 (1M)	14 (12F, 2M)
	**FGD**	Social Media Influencers, n = 4 (4F) Mahalla Representatives, n = 2 (1F, 1M)		0	Mahalla Representatives, n = 2 (2F)	
**National Level Key Informants**	**IDI**	Health Ministry Officials, n = 4 (4F) Health Ministry Specialists, n = 1 (1F) Education Ministry Official, n = 1 (1M) Women’s Committee Official, n = 1 (1F)				17 (10F, 7M)
	**FGD**	Ministry Specialists (Oncology), n = 3 (1F, 2M) Ministry Specialists (Gynaecology), n = 2 (1F, 1M) Muslim Board Officials, n = 5 (2F, 3M)				

Note: For each participant group, the breakdown of female (F) and male (M) participants is given in parentheses after the total number of participants.

## Data Availability

Data in the form of anonymized transcripts are available on request from S.M.N. until 2024 when they will be destroyed as per conditions of ethical approval.

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
