# Peer review of "Formative Research on HPV Vaccine Acceptance among Health Workers, Teachers, Parents, and Social Influencers in Uzbekistan"

_vaccines, 2023, doi:10.3390/vaccines11040754_

Round 1

Reviewer 1 Report

To my opinion, the presented research work is relevant to the topic of the Special Issue of Vaccines, i.e. HPV Vaccination: Current Situation and Future Goals

In many high-income countries, the implementation of HPV vaccination programs has led to a substantial reduction in HPV-induced cancers. However, the situation is not that optimistic for low- and middle-income countries and the reasons for that are mainly the low immunization coverage and belated introduction of national HPV immunization programs. Pilot studies in these countries concerning issues related to general knowledge about the vaccine among population groups, attitude to vaccination, acceptance, hesitancy, etc. are welcome.

In fact, I have no major comments on the submitted manuscript. Below follow some minor remarks:

Minor remarks:

1. Introduction, lines 33-38 – I would suggest the introductory statements be supported with relevant references.

2. Table 2 – I would suggest abbreviations F and M be explained as Male and Female below the table.

3. Results, line 175DTP should be written in full before introducing the abbreviation, albeit very popular.

4. Results, lines 181, 212, 227, Discussion, line 338 – an abbreviation for Ministry of Health was already introduced on line 47. Please use the abbreviation MoH in the text following line 47.

5. Discussion, lines 295-298. I consider the text on these lines not relevant.

Author Response

Response to Comments: Reviewer 1

Dear Reviewer 1,

  Thank you for your time, encouraging comments, and instructive remarks. Following comments of both reviewers and the editor, we have made several changes to the manuscript. Please find below in blue responses to your remarks.

With best regards,

Sahil Warsi

  1. Introduction, lines 33-38 – I would suggest the introductory statements be supported with relevant references.

Thank you for alerting us to this. The citation field at the end of the paragraph was somehow blank. We have now inserted the appropriate references throughout the paragraph.

  1. Table 2 – I would suggest abbreviations F and M be explained as Male and Female below the table.

We have added the following note in the footer to table 2 following your suggestion: “For each participant group, the breakdown of female (F) and male (M) participants is given in parentheses after the total number of participants.”

  1. Results, line 175DTP should be written in full before introducing the abbreviation, albeit very popular.

Considering the flow of the text, we have replaced “DTP” with “Diphtheria-Tetanus-Pertussis” in brackets, so that the quotation now reads:  “There were some reactions after the [Diphtheria-Tetanus-Pertussis] vaccine, such as a fever… polio leads to paralysis… since [HPV vaccine is] against cancer, would it cause cancer?"

  1. Results, lines 181, 212, 227, Discussion, line 338 – an abbreviation for Ministry of Health was already introduced on line 47. Please use the abbreviation MoH in the text following line 47.

Thank you again for pointing this out. We have replaced all instances of “Ministry of Health” after line 47 with “MoH”, except for in the comment on Funding at the end of the article.

  1. Discussion, lines 295-298. I consider the text on these lines not relevant

Again, thank you very much for raising this issue. This was boiler-plate text from the template that did not get deleted. It has been removed.

Reviewer 2 Report

I have reviewed the paper by Warsi et al.,

The abstract basically omits mentioning actual results. It jumps from the purpose of using the collected data to kind of a conclusion list.

Details on sample size determination will 97 be discussed in a future publication.”…Why? Why not here where it is needed. The number of participants is rather small, so the issue of the sample size is particularly important to be explained.

Dialogues could be provided as a supplementary table or file, and not in the narrative of the manuscript.

There is a lot of crunched citations in the Introduction, and at the same time a lack of citations when they are actually needed. For example

Participants across all groups indicated some parents might refuse specific vaccines like the pentavalent vaccine due to misinformation on potential side effects, though key informants from the ministry of health suggested misinformation on vaccination was more prevalent in among urban parents.” No citations at all. Why misinformation? Couldn’t these be genuine concerns on side effects? What is the basis for the government to call something misinformation, is unknown since no citations were provided. The misinformation issue, is surprisingly left out completely from the Discussion, despite its importance.

Author Response

Response to Comments: Reviewer 2

Dear Reviewer 2,

  Thank you for your time and remark. Following comments of both reviewers and the editor, we have made several changes to the manuscript. Please find below, in blue, responses to your remarks. While comments were not initially numbered, we have numbered them according to how they fit together.

Best regards,

Sahil Warsi

  1. The abstract basically omits mentioning actual results. It jumps from the purpose of using the collected data to kind of a conclusion list.

To clarify the COM-B findings, which might have appeared as conclusions, we have amended the final part of the abstract as follows: “Capability findings indicated participants understood cervical cancer was a national health issue, but HPV and HPV vaccine knowledge was limited among non-health professionals, some nurses, and rural health workers. Results on opportunity for accepting HPV vaccine sowed most participants would do so if they had access to credible information on vaccine safety and evidence. Regarding motivation, all participant groups voiced concern about potential effects on young girls’ future fertility. Echoing global research, study results highlighted how trust in health workers and government as health-related information sources, and collaboration among schools, municipalities, and polyclinics could support potential vaccine acceptance and uptake”

  1. “Details on sample size determination wil be discussed in a future publication.”…Why? Why not here where it is needed. The number of participants is rather small, so the issue of the sample size is particularly important to be explained..

A second publication on the process of using theory-based rapid qualitative research to support HPV vaccine introduction in an LMIC contexts has already been prepared and is being updated in light of recent work in another country in the region. This study’s sample size is considerable for this kind of applied qualitative formative research, and the other article considers in more detail the factors to consider in thinking through information power, available resources, and sample size in such cases.

The comment alerted us to what was missing. The methods section has now be restructured to include a first section delineating the process of study design, including what is considered under the concept of information power for qualitative research in order to determine sample size. Given further details on design and sample size are now provided, the statement that details are discussed in a future publication has been removed.

  1. Dialogues could be provided as a supplementary table or file, and not in the narrative of the manuscript. There is a lot of crunched citations in the Introduction, and at the same time a lack of citations when they are actually needed.

As this is a qualitative study based on thematic analysis, the quotations are illustrative of the points raised in the body, and are perhaps the citations suggested to be missing. We have thus added a paragraph to the methods section on the analysis process that outlines how data was analyzed and how the example quotations are connected to the findings.

Including quotations in the manuscript is a convention for reporting on such qualitative public health research, and after consideration we have kept the quotations in the text. However, the comment alerted us to the fact formatting of the quotations was identical to the body paragraphs, which might have contributed to unclarity on the purpose of quotations. We have now increased the indentation and justified the quotations text so that it is visually clear they are illustrative examples relating to the preceding paragraph.

Since transcripts are written verbatim, it is sometimes difficult to find quotations that can be presented as whole comprehensible sentences, even if the idea is expressed. This is because, while speaking, people often leave sentences half finished, switch topics, or perhaps tangentially mention an issue only to elaborate on it later in a way that can’t be visually combined. While quotations cannot be provided for every point, the data and analysis is available, as specified in the data availability statement.

  1. For example: “Participants across all groups indicated some parents might refuse specific vaccines like the pentavalent vaccine due to misinformation on potential side effects, though key informants from the ministry of health suggested misinformation on vaccination was more prevalent in among urban parents.” No citations at all. Why misinformation? Couldn’t these be genuine concerns on side effects? What is the basis for the government to call something misinformation, is unknown since no citations were provided.

Thank you for raising this point, it drew our attention to two issues. First, the topic sentence of the paragraph was missing. The main point of the paragraph was to indicate a difference in hesitancy between urban and rural parents, and we have added “Data suggested a difference in urban and rural parents’ level of vaccine hesitancy, though misinformation, unexpectedly, did not appear to be a widespread issue..” The sentences following the ones indicated in remark 4 above highlight how the opinion of MoH representatives can be triangulated with responses from teachers and health workers that urban parents’ vaccine hesitancy is higher than rural parents’.

The subsequent paragraphs build on this point to illustrate hesitancy is connected to safety concerns and knowledge gaps in different target groups. This relates to the question raised in the remark on side effects. Most people were not concerned about side effects of vaccination, other than that on girls’ fertility, which is discussed under motivation-factor findings.

The second issue this remark alerted us to, was that in this instance and one other instance under opportunity findings, the frequency of the finding was not indicated. We have clarified that it was just some participants across all target groups who suggested misinformation might be an issue. The change under opportunity findings was to indicate that a majority of participants across target groups wanted information from health workers.

We also appreciate that even though misinformation was not raised as a major issue, what this meant contextually was not clear. We have now included the following, quotation of an MoH official from Mother and Child Care: “…[urban mothers share]  negative information on Telegram … They have their own groups for mothers and they throw in the information to exchange with each other: somewhere, someone got sick, somewhere, someone has died from vaccination, etc… I believe that incorrect information is the main reason for refusing from vaccination.”

  1. The misinformation issue, is surprisingly left out completely from the Discussion, despite its importance.

Misinformation was included in the discussion in lines 317-322. As explained, unlike in other contexts, misinformation was not a significant issue in Uzbekistan in 2019 outside the capital and this was outlined in the discussion. The original discussion paragraph (lines 314-322) read as follows:

“Research participants across all groups were found to be generally supportive of the new HPV vaccine introduction, though a majority highlighted the need for clear, accessible, and credible information on the evidence base and safety of the HPV vaccine in order to be confident in their support for an HPV vaccine. Misinformation on vaccines did not appear to be widespread among research participants, though there was some indication of vaccine hesitancy among urban parents and teachers, possibly spread via social media. This is not surprising, considering that internet coverage and access remains out of reach for many in the country outside the capital, and it was only after fieldwork was conducted that many social media sites were unblocked in the country.(35,36).”

We have now edited the paragraph (lines 665-670) with an additional sentence contrasting the 2019 situation in Uzbekistan with other HIC/LMIC contexts where misinformation was a significant issue:

Research participants across all groups were found to be generally supportive of the new HPV vaccine introduction, though a majority highlighted the need for clear, accessible, and credible information on the evidence base and safety of the HPV vaccine in order to be confident in their support for an HPV vaccine. Misinformation on vaccines did not appear to be widespread among research participants, though there was some indication of vaccine hesitancy among urban parents and teachers, possibly spread via social media. This situation differed markedly from other high income and LMIC contexts, where even before the COVID-19 pandemic, misinformation through media contributed to HPV vaccine hesitancy, decreased uptake, or even government suspension of HPV vaccination.[36–39] The finding of low levels of misinformation is not surprising, considering that, at the time of research, internet coverage and access remained out of reach for many in the country outside the capital, and it was only after fieldwork ended that many social media sites were unblocked across the country.[40,41]

Round 2

Reviewer 2 Report

The directionality of the paper is basically to contrast urban vs. rural in the context of HPV vaccine acceptability.

Main table 2 shows this (Nothing else close to "data" is shown anywhere else). Desite this, it doesn't seem to be the tone of the paper (even from the title).

I suggest merging this with your manuscript in preparation that contains the larger data that you are mentioning, and make this a more thorough and sound paper. 

Author Response

Dear Reviewer,

  We appreciate your time in reviewing the article, have read and discussed the comments, and provide our response in blue below.

With thanks,

Sahil Warsi

The directionality of the paper is basically to contrast urban vs. rural in the context of HPV vaccine acceptability.

The aim of the article was to report on COM-B findings for behaviour affecting HPV vaccine acceptance among key target groups (health workers, teachers, parents, and social influencers). Data was compared across target groups, within target groups (age, sex, social background as available), and study sites representing multiple, differing urban and rural contexts (lines 108-115). The urban-rural divide pertained to some findings, but most themes/findings cut across groups.

Main table 2 shows this (Nothing else close to "data" is shown anywhere else). Desite this, it doesn't seem to be the tone of the paper (even from the title).

Table 2 is not study data; it is an overview of participants and research activities across field sites. As this was a qualitative study, the data are the words, statements, and ideas expressed in the transcripts of focus group discussions and interviews. The quotations provided are exemplary and reflect this data, as described in lines 370-379.

I suggest merging this with your manuscript in preparation that contains the larger data that you are mentioning, and make this a more thorough and sound paper. 

After discussion, we have decided to keep both articles separate as results and process papers. The data in the form of quotations is provided in this article. The second article is a more in-depth reflection on the process and lessons learned for conducting such applied, qualitative research to inform public health interventions on HPV vaccine introduction in low resource settings.